# Dissociable dynamic effects of expectation during statistical learning

Hannah H McDermott[1,2,3]*, Federico de Martino[3], Caspar M Schwiedrzik[4,5,6], Ryszard Auksztulewicz[1,3]

[1]Department of Education and Psychology, Freie Universität Berlin, Berlin, Germany; [2]Berlin School of Mind and Brain, Berlin, Germany; [3]Faculty of Psychology and Neuroscience, Maastricht University, Maastricht, Netherlands; [4]Neural Circuits and Cognition Lab, European Neuroscience Institute Göttingen—A Joint Initiative of the University Medical Center Göttingen and the Max Planck Institute for Multidisciplinary Sciences, Göttingen, Germany; [5]Perception and Plasticity Group, German Primate Center, Leibniz Institute for Primate Research, Göttingen, Germany; [6]Cognitive Neurobiology, Research Center One Health Ruhr, University Alliance Ruhr, Faculty of Biology and Biotechnology, Ruhr-University Bochum, Bochum, Germany

*For correspondence:
h.mcdermott@fu-berlin.de

## eLife Assessment

This **important** study is of relevance to the fields of predictive processing, perception, and learning, with a well-designed paradigm allowing the authors to avoid several common confounds in investigating predictions, such as adaptation. Using a state-of-the-art multivariate EEG approach, the authors test the opposing process theory and find evidence in support of it. Overall, the empirical evidence is **solid**; however, some conclusions rest on limited evidence and need further work to reconcile the present results with previous studies.

**Abstract** The brain is thought to optimise behaviour by generating predictions based on learned statistical regularities. Predictive processing seemingly explains expectation suppression (ES), the attenuation of neural activity in response to expected stimuli. However, the mechanisms behind ES are unclear, with conflicting evidence for alternative models. Sharpening models propose that expectations suppress neurons away from the expected stimulus, increasing the signal-to-noise ratio and boosting decoding for expected stimuli. In contrast, dampening models posit that expectations suppress neurons that are tuned to the expected stimuli, reducing overall response magnitude and decoding accuracy. The opposing process theory (OPT) suggests that both processes occur at different time points, namely that initial sharpening is followed by later dampening of the neural representations of the expected stimulus. Here we test this theory and shed light on the dynamics of expectation effects, both at single-trial level and over time. Thirty-one participants completed a statistical learning task in which a 'leading' image from one category predicted a 'trailing' image from a different category. Multivariate EEG analyses decoded stimulus information related to the trailing category. Within-trial, expectation increased decoding accuracy at early latencies and decreased it at later latencies, in line with OPT. However, across trials, stimulus expectation decreased decoding accuracy in initial trials and increased it in later trials. We theorise that these dissociable dynamics of expectation effects within and across trials support hierarchical learning mechanisms. While within-trial results support the OPT, across-trial results suggest that sharpening and dampening effects emerge at distinct stages of associative learning.

## Introduction

Influential theories of perception and learning suggest a key role of prediction in both processes (*Friston, 2005*; *Jiang and Rao, 2024*; *Walsh et al., 2020*). However, it remains unclear whether the influence of prediction on neural activity across sources of prediction, time scales and learning stages can be explained by a single mechanism (*Blom et al., 2021*; *Malekshahi et al., 2016*; *Wang et al., 2017*; *Wischnewski and Schutter, 2019*). Repetition suppression (RS), or the attenuation of neural activity in response to repeated (and therefore predictable) stimuli, has been well documented across a range of sensory modalities, species, and experimental paradigms (*Auksztulewicz and Friston, 2016*; *Grotheer and Kovács, 2016*). Predictive processing theories explain RS as a manifestation of minimising prediction errors (PEs) through adaptive changes in predictions about the content and precision of sensory inputs (*Auksztulewicz and Friston, 2016*). Expectation suppression (ES), or the attenuation of neural activity in response to expected stimuli (regardless of its frequency), appears to be a related phenomenon, but ES is more controversial in terms of its underlying mechanisms. In an earlier study attempting at dissociating the two phenomena, it has been shown that RS and ES affect neural activity in separable time windows (*Todorovic and de Lange, 2012*). More recently, however, *Feuerriegel et al., 2021* pointed out major challenges to the evidence underlying ES in the visual system, such as the conflation of ES with RS, surprise effects, attention effects, and stimulus novelty. While empirical evidence for ES is more fragmentary (*Feuerriegel et al., 2021*), it is in principle a more convincing neural signature of predictive processing, as it is less readily explained than RS by mere neural adaptation (*Barron et al., 2016*), which does not necessarily reflect cognitive processes.

At least two types of neural activity modulation have been linked to the effects of expectation. Sharpening models propose that expectations suppress neurons that are not tuned to the expected stimulus, thus inhibiting information inconsistent with the expectation (*Kok et al., 2012*; *Yon et al., 2018*). Depending on the level of analysis, this may result in sharper tuning curves (i.e., increased signal-to-noise ratio for expected stimuli) for a particular neuron or small population, amounting to relative expectation enhancement; and/or in sparse responses (i.e., net activity decrease) in a larger population (*Kremláček et al., 2016*). In contrast, dampening models posit that expectations suppress neurons that are tuned to the expected stimuli, effectively 'explaining away' the prediction error responses (*Kumar et al., 2017*). By cancelling information in line with expectations, uninformative activity in the sensory stream is prevented while favouring novel information (*Meyer and Olson, 2011*; *Richter et al., 2018*). Studies attempting to settle the debate have produced mixed results on both accounts. *Kok et al., 2012* and Garlichs & Blank (*Garlichs and Blank, 2024*) both found evidence for neural sharpening in the visual domain using fMRI and multivariate pattern analysis (MVPA). Another study using 7T fMRI found both suppression of irrelevant neural responses in visual areas and selective enhancement of relevant neural responses in higher-order frontoparietal cortices (*González-García and He, 2021*). In contrast, *Richter et al., 2018* and *Richter et al., 2022* demonstrated evidence in favour of dampening of neural responses in the visual domain. In the latter study, after training participants on a statistical learning task, stimulus features were decoded from fMRI signals separately in expected vs. unexpected trials. The rationale of the analysis was that, if expectation leads to dampening, the response amplitudes will be lower, and as a result stimulus decoding should be lower in expected trials; conversely, if expectation leads to sharpening, the signal-to-noise ratio will increase, and as a result stimulus decoding should increase in expected trials. Both studies found lower decoding for expected trials, consistent with suppressed activity of neuronal populations tuned towards expected stimuli. However, electrophysiological research in the area is scarce (*Meyer and Olson, 2011*; *Schwiedrzik and Freiwald, 2017*), despite its potential for parsing out temporal differences which may go unnoticed in fMRI studies with poor temporal resolution.

Taking temporal differences into account, the recently suggested OPT has suggested that sharpening and dampening may both occur, but at different time points during the predictive process (*Press et al., 2020*). This model seeks to reconcile contrasting findings as some studies show that expected events are perceived with greater intensity 50 ms after presentation, but this bias reverses around 200 ms (*Press et al., 2010*; *Yon and Press, 2017*), while classically reported ES effects most often take place ~150 ms after presentation (*Hughes et al., 2013*). OPT suggests that within each trial, initial processing relies on prior knowledge to sharpen sensory representations towards the expected stimuli, and a later processing stage follows, dampening the neural representations of the expected stimulus. This is based in part on trends in prior research. Studies have found that while expected

events are perceived with greater intensity 50 ms after presentation, this bias reverses by 200 ms such that unexpected events are perceived with greater intensity (*Press et al., 2010*; *Yon and Press, 2017*). While this theory is largely untested thus far, it will prove a highly useful benchmark for future research. Namely, by implementing decoding analyses in a time-resolved manner, we can better parse out the intricacies of this model. Sharpening and dampening models have previously been investigated using time-resolved methods, albeit with different study designs. *Han et al., 2019* found that stimulus-specific event-related activity is dampened for expected tones in the auditory cortex, while studies examining the inferior temporal (IT) cortex in monkeys have found that IT neurons represent less accurately predictable compared with unpredictable stimuli (*Kumar et al., 2017*; *Meyer and Olson, 2011*). It is worth noting that while the above studies did change the pairings of stimuli between trials, they used small pools of stimuli (<30) which were repeated many times and implemented extensive training and exposure to the stimuli prior to collecting data.

Theoretical work has proposed that, similar to the effects of prediction within a trial, learning should also be linked to neural activity suppression, albeit at hierarchically higher levels of neural processing (*Kiebel et al., 2008*). However, empirical work yielded more nuanced results. Previous research has focused on the mechanisms of ES after learning has taken place, while the effects of expectation across trials (i.e., during learning) have been largely overlooked. In an earlier fMRI study (*Müller et al., 2013*), it has been shown that repetition effects are qualitatively different for the initial vs. later stimulus repetitions, gradually enhancing vs. suppressing BOLD activity, respectively. In contrast, EEG work has shown that neural responses are more strongly modulated by initially formed repetition-based predictions than by their subsequent revisions (*Fitzgerald et al., 2021*). Furthermore, in an MEG study differentiating the effects of expectation and familiarity, both factors were linked to suppressed neural activity but expectation (previously seen stimuli with expected vs. unexpected structure) attenuated activity in the lateral occipital cortex, whereas familiarity (previously seen vs. unseen stimuli with no predictable structure) additionally led to temporal sharpening of evoked responses in early visual regions (*Manahova et al., 2018*). However, the study only characterised these effects after learning, averaging across many trials. While theoretical accounts such as the reverse hierarchy theory of perceptual learning (*Ahissar and Hochstein, 2004*) and hierarchical Bayesian framework of statistical learning (*Fiser and Lengyel, 2019*) suggest that perceptual learning effects progress from higher to lower levels of the visual system, in classical predictive coding hierarchies the time scales increase along the processing hierarchy in an ascending manner (*Kiebel et al., 2008*). Thus, it is unclear whether dampening vs. sharpening effects (previously found to occur in anterior vs. posterior regions) (*González-García and He, 2021*) should be identified at earlier vs. later learning stages.

Here, we assessed the effects of expectation on EEG-based stimulus decoding during statistical learning while controlling for repetition effects, which may render the expectation effects impossible to parse out given the similarity between the two phenomena. We focused on empirically testing the OPT on visually evoked responses. We expected to find univariate evidence of expectation suppression, by way of reduced ERP amplitude to trailing images in valid trials over occipital channels. Furthermore, in line with the OPT, we hypothesised that sharpening effects will lead to higher decoding accuracy for valid trailing images earlier within trials, and dampening effects will lead to higher decoding for invalid images later within trials. We also explored the dynamics of expectation effects across trials. In brief, our results show support for OPT, however, with different dynamics within trials than across learning stages.

## Results

Participants (N=31) were exposed to a sequence of visual scenes (*Figure 1A*) while their neural activity was recorded using EEG (see 'Methods' for more details). The sequence was manipulated with respect to the expectation of image categories. In each trial, participants were presented with two images from nine different categories in quick succession, with five 'Leading' categories, and four 'Trailing' categories. Over 3500 categorised images were used to avoid the repetition of individual stimuli and control for RS. Category pairs and the transitional probabilities between them were determined by the transitional probability matrix depicted in *Figure 1B*. Each participant viewed 1728 trials without prior training, and each image was only presented once.

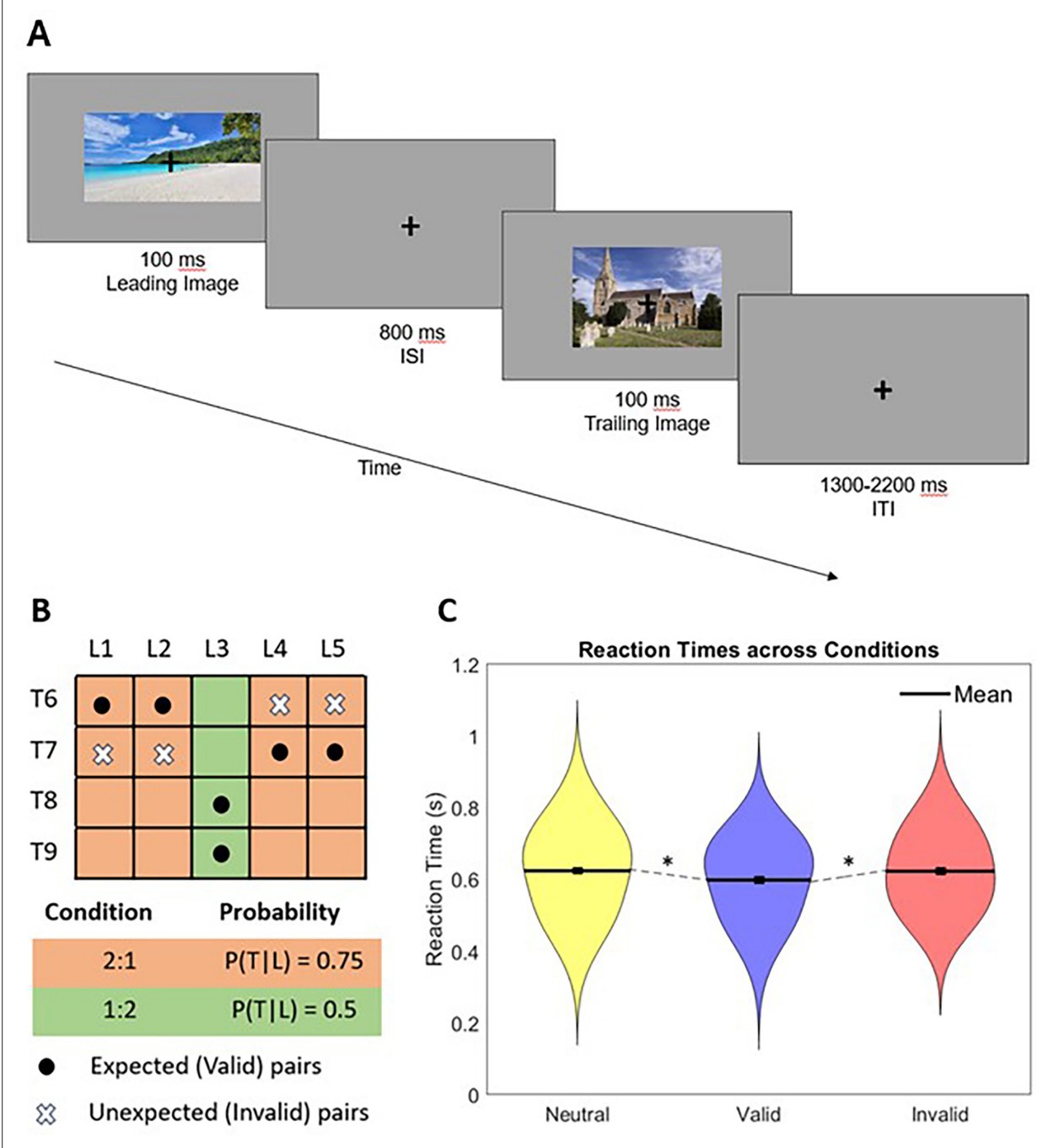

**Figure 1.** Paradigm overview. (**A**) A single trial, with two example images. Images were presented for 100 ms each with 800 ms interstimulus interval and an intertrial interval of 1300–2200 ms. Participants were required to respond to upside down images with button press; all upside down images were trailing. (**B**) Transitional probability matrix determining category pairs. Five leading categories and four trailing categories were used. Orange represents the 2:1 condition, and green represents the 1:2 (control) condition. Cells with dots represent the valid pairs, cells with Xs represent the invalid pairs, and empty cells represent non-existent pairs. (**C**) Reaction times across Valid, Invalid, and Neutral conditions. Asterisks indicate significant results at p<0.05 after correction for multiple comparisons.

## Behavioural results

After the EEG recording, a majority of participants (N=21) were asked to perform a categorisation task on images drawn from the same categories. Transitional probabilities in this task were kept the same as in the EEG session, and participants performed a speeded indoor/outdoor categorisation task on the 'Trailing' images. A repeated-measures ANOVA was conducted to compare reaction times (RTs)

across conditions in order to assess implicit learning, which showed significant differences between conditions ($F_{(2,40)}$ = 4.231, p=0.022, $\eta^2$=0.175). Paired t-tests were conducted to further elucidate RT differences between conditions. Participants reacted significantly faster in valid trials (mean = 598.9 ms, SEM = 7.2 ms) than in invalid trials (mean = 623.6 ms, SEM = 7.2 ms; $t_{20}$=–2.18, p=0.041, d=0.476, two-tailed). Participants also reacted significantly faster in valid trials than in neutral trials (mean = 624.7 ms, SEM = 5.7 ms; $t_{20}$=2.88, p=0.00, d=0.629, two-tailed). There was no significant difference in RTs between invalid and neutral trials ($t_{20}$=0.120, p=0.906, d=0.026, two-tailed). Significant differences between both valid and invalid, and valid and neutral conditions suggest that associations between pairs were learned implicitly (*Figure 1C*).

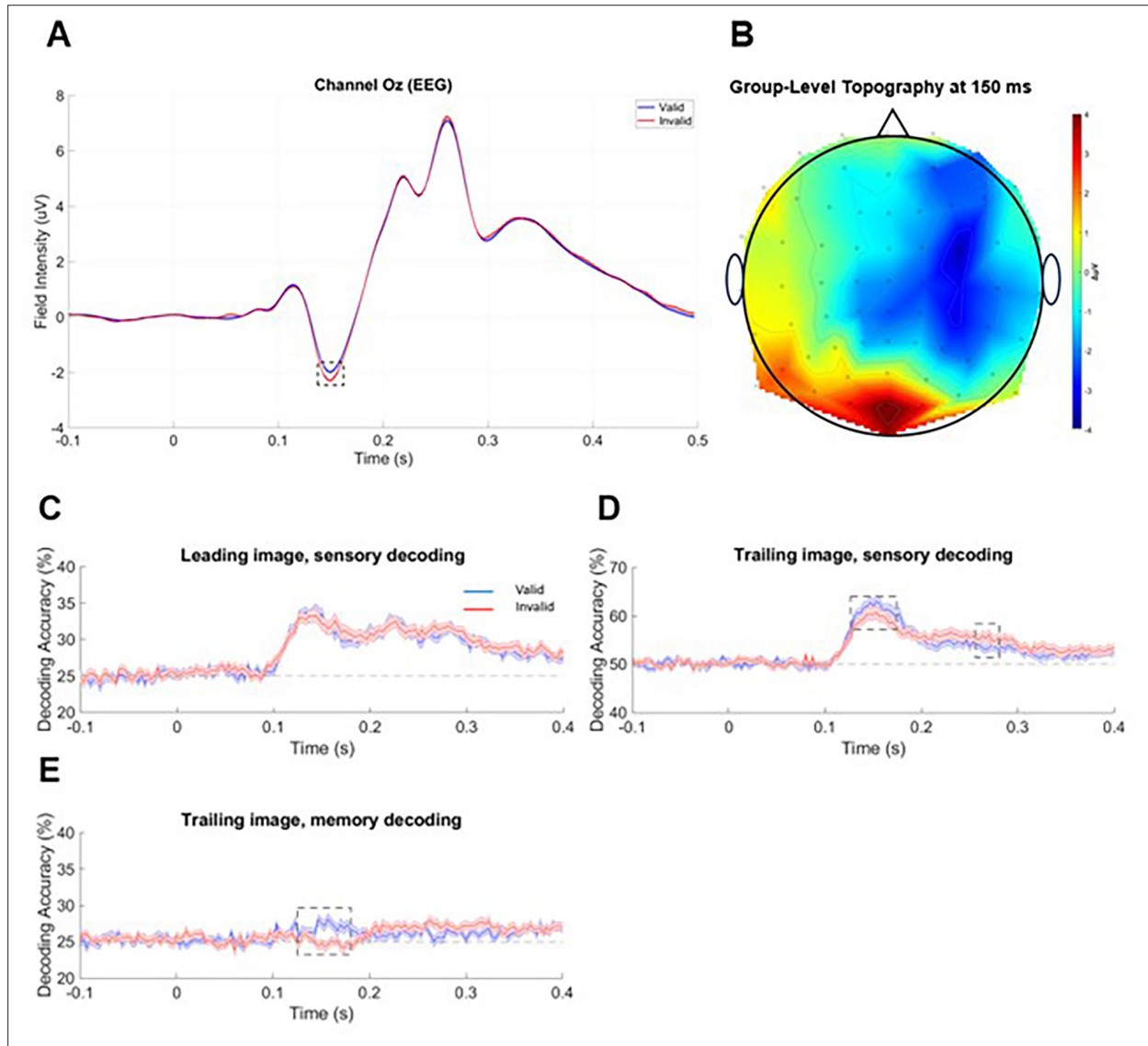

**Figure 2.** Results of data analysis. (**A**) Grand average ERP statistical analyses in a cluster of occipital channels at Channel Oz. Significant difference between valid and invalid trial amplitudes. (**B**) Topography plot of the t-values resulting from paired t-tests comparing Valid and Invalid EEG data at 150 ms. (**C**) SVM decoding results for Leading image sensory decoding. Dashed vertical line at 0 s indicates stimulus onset. (**D**) Decoding results for Trailing image sensory decoding. Dashed rectangles denote latencies of significant effects ($p_{FWE}$<0.05). Dashed vertical line at 0 s indicates stimulus onset. (**E**) Decoding results for Trailing image memory decoding. Dashed rectangle denotes latencies of significant effects ($p_{FWE}$<0.05). Dashed vertical line at 0 s indicates stimulus onset.

## Univariate EEG analyses

In order to assess the impact of expected vs. unexpected trials on visually evoked ERPs, a paired t-test was conducted on the epoched amplitudes in a cluster of occipital channels (Oz, O1, O2) at channel Oz. Since previous research has questioned the robustness of univariate effects of ES (*Feuerriegel et al., 2021*), we conducted a target region-of-interest analysis on the channel showing peak amplitude of the visually evoked response (*Figure 2B*). Unexpected trials resulted in a significantly more pronounced negative deflection 150 ms after image onset ($t_{2926}=3.3789$, $p \leq 0.00123$) after FWE correction (*Figure 2A*). This is in line with the idea that neural responses to expected stimuli are attenuated in ES. A more conservative and exploratory whole-brain analysis, correcting for all channels and time points, failed to reach significance, consistent with the issues raised by *Feuerriegel et al., 2021*. The group-level topography of the amplitudes differences between valid and invalid trials is represented in *Figure 2B*.

## Decoding analyses

To test the hypothesis derived from the OPT that expectation should have opposing effects at earlier (sharpening) vs. later (dampening) response latencies, we used multivariate decoding analysis. Following previous work (*Richter et al., 2022*; *Han et al., 2019*), we reasoned that sharpening should lead to increased decoding accuracy for expected stimuli, while dampening should lead to increased decoding accuracy for unexpected stimuli. To this end, we calculated a linear SVM-based category decoding accuracy at each time point based on EEG amplitudes, separately for valid vs. invalid trials (*Han et al., 2019*). Three separate decoding analyses were conducted: 'sensory decoding' (decoding the visual category of the presented image, done separately for the leading and trailing images), and 'memory decoding' (decoding the preceding visual category based on the trailing images). The above analyses were all subject to cluster-based family-wise error (FWE) correction for multiple comparisons across time points. FWE is a standard method of correcting for multiple comparisons and decreasing the cluster-based false-positive ratio implemented in SPM. It uses random field theory assumptions (*Kilner et al., 2005*) to account for the spatiotemporal correlation between neighbouring data points. Clusters were thresholded to 0.05 uncorrected, then reported clusters which were significant at cluster-level $p_{FWE}<0.05$.

In the sensory decoding analysis based on leading images, decoding accuracy was above chance for both valid ($T_{max}=4.79$, $p_{FWE}<0.001$) and invalid trials ($T_{max}=4.02$, $p_{FWE}<0.001$) from 100 ms, with no significant difference between them ($T_{max}=2.47$, $p_{FWE}=0.987$) (*Figure 2C*). This initial analysis largely served as a sanity check since no difference between decoding the category of valid and invalid images should be observed as the leading image cannot be invalid.

On the contrary, in sensory decoding of trailing images, decoding accuracy was significantly different between valid and invalid images 123-180 ms after image onset, where decoding the category of valid images was significantly higher than invalid images (cluster-level statistics: $T_{max}=3.62$, $p_{FWE}<0.001$), but decoding the category of invalid images was significantly higher than valid images 280–296 ms after image onset ($T_{max}=3.89$, $p_{FWE}<0.001$) (*Figure 2D*). Therefore, stimulus category expectation had opposing effects on trailing category decoding based on EEG signals early vs. late within a trial. The decoding accuracy of both valid ($T_{max}=5.10$, $p_{FWE}<0.001$) and invalid trials ($T_{max}=3.83$, $p_{FWE}<0.001$) was also above chance in this condition from 100 ms.

Finally, as with sensory decoding of trailing images, memory decoding accuracy based on valid trailing images was significantly higher than invalid trailing images 123-180 ms after image onset ($T_{max}=3.96$, $p_{FWE}<0.001$) (*Figure 2E*). Both valid ($T_{max}=4.67$, $p_{FWE}=0.047$; from 100 ms) and invalid ($T_{max}=3.40$, $p_{FWE}<0.001$; from 200 ms) trailing images yielded above-chance decoding.

In order to better assess how the difference in prediction and memory decoding between valid and invalid pairs changes during learning, we separated each data set into four trial bins. In order to assess the utility of examining each bin in more detail, we calculated the trial-by-trial time-series of the decoding accuracy benefit for valid vs. invalid for each participant and averaged this benefit across time points for each of the two significant time windows. Based on this, we fitted a logarithmic model to quantify the change of this benefit over trials, then found the trial index for which the change of the logarithmic fit was <0.1%, indicating stabilised decoding accuracy. This demonstrates that expectation effects developed rapidly early in learning and stabilised approximately after 50% of trials, after

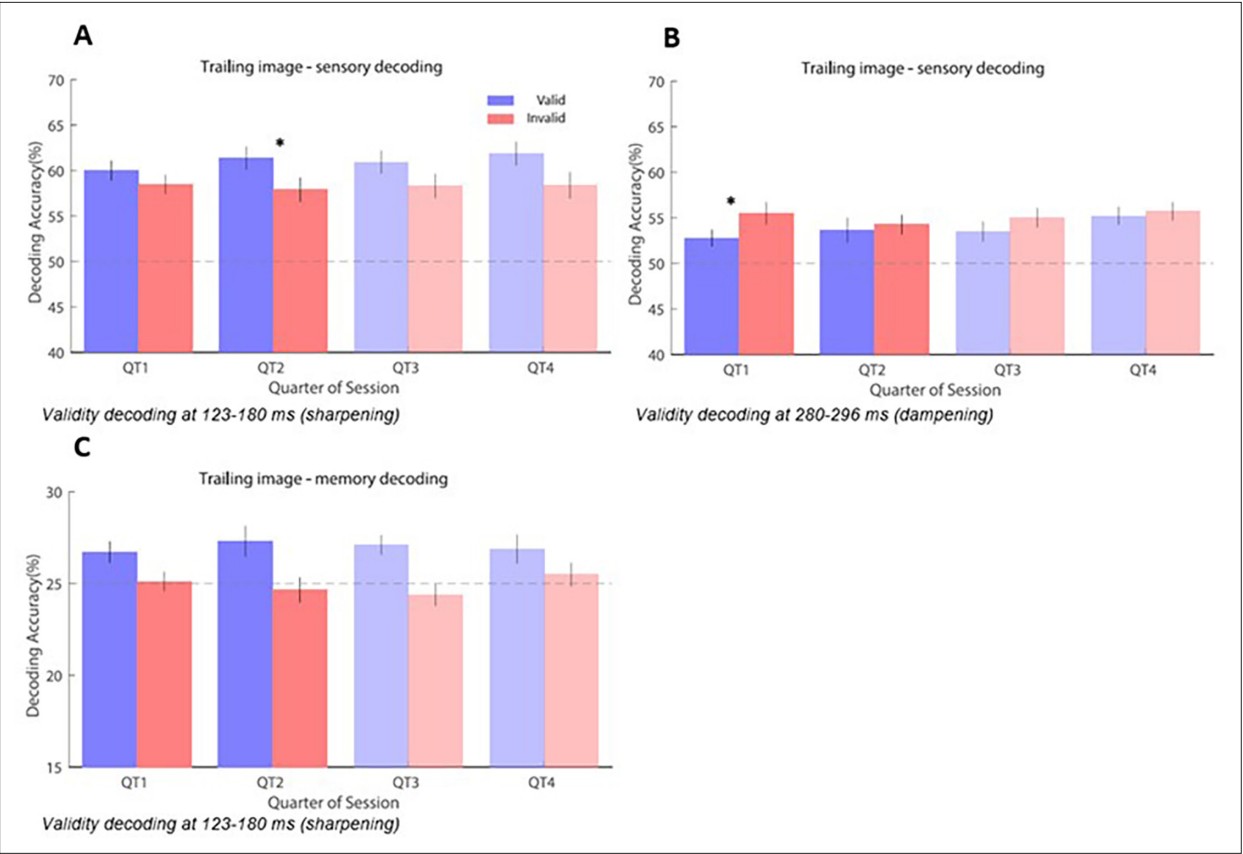

**Figure 3.** Analysis of validity effects on decoding over trial bins. Error bars indicate SEM. Dashed horizontal lines indicate chance level. Asterisks indicate significant results at p<0.05 after correction for multiple comparisons. (**A**) Learning over time at 123–180 ms in sensory decoding. (**B**) Learning over time at 280–196 ms in sensory decoding. (**C**) Learning over time at 280–296 ms in memory decoding.

which almost zero learning took place. Given the results of this analysis and to ensure a sufficient number of trials, we focussed our further analyses on bins 1–2 to directly assess the effects of learning.

We separately analysed the early (123–180 ms) and late (280–296 ms) within-trial time window identified in the main analysis of sensory decoding based on the trailing images (*Figure 2D*) as well as the early (123–180 ms) time window identified in the memory decoding analysis (*Figure 2E*). We fit each of these effects to a linear mixed effects model with Holm–Bonferroni correction for multiple comparisons across bins. Outliers larger than 3 SDs based on Cook's distance were removed.

In the early time window of sensory decoding, the main effects of validity ($T_{79}$=0.84141, p=0.403) and bin ($T_{79}$=1.9242, p=0.058) were not significant. The interaction between validity and bin was significant, however ($T_{79}$=–2.1679, p=0.033). Cohen's f for the full model was f=1.688. Post hoc analyses of the effects of each bin showed no significant differences between valid and invalid trials in the first bin ($T_{28}$=1.2036, p=0.239, f=0.052) but significant differences in the second bin ($T_{23}$=3.8704, p<0.001, f=0.651) (*Figure 3A*). In other words, the decoding benefit of valid predictions—visible at early post-stimulus latencies when averaging over all trials—took approximately 15 minutes (or 50 expected trials per image pairing) to become significant. Additional post hoc analyses comparing the first bin and the second bin found significant differences in invalid trials ($T_{24}$=–2.208, p=0.037, f=0.203), but not in valid trials ($T_{26}$=1.449, p=0.159, f=0.081). Outliers larger than 3 SDs based on Cook's distance were removed prior to this analysis.

Analyses of the later peak in sensory decoding found the opposite effect. The main effect of validity ($T_{77}$=3.5402, p<0.001), and the main effect of bin ($T_{77}$=2.2399, p=0.028) both contributed significantly. The interaction between them ($T_{77}$=–2.6854, p=0.009) also showed significant differences. Cohen's f for the full model was f=1.5943. In contrast to the early peak, post hoc analyses showed that the first bin had significant differences between valid and invalid trials ($T_{24}$=4.293, p<0.001, f=0.768) but the second bin did not ($T_{24}$=0.2673, p=0.791, f=0.003) (*Figure 3B*). In other words, the decoding benefit of

invalid predictions—visible at late post-stimulus latencies when averaging over all trials—disappeared after approximately 15 minutes (or 50 expected trials per image pairing). As with the early peak, additional post hoc analyses comparing the first bin and the second bin found significant differences in invalid trials ($T_{26}$=–2.607, p=0.015, f=0.261), but not in valid trials ($T_{22}$=1.273, p=0.216, f=0.074). Outliers larger than 3 SDs based on Cook's distance were removed prior to this analysis.

Finally, in the memory decoding, each of the main effects for validity ($T_{1,77}$=-0.407, p=0.685), bin ($T_{1,77}$=0.751, p=0.455), and the interaction between them ($T_{1,77}$=-0.794, p=0.429) failed to find significant effects after removal of outliers (*Figure 3C*). Cohen's f for the full model was f=0.997.

In order to examine the extent to which participants' neural responses were linked to their behavioural responses, we calculated Pearson correlation coefficients for behavioural responses (difference in RTs), neural responses (difference in decoding accuracy) during the sharpening effect, and neural responses during the dampening effect based on the difference in mean responses for each participant. Correlations between the behavioural responses and the sharpening effect ($r_{19}$=0.121, p=0.602) and between the behavioural responses and the dampening effect ($r_{19}$=0.127, p=0.584) were not significant. It should be noted, however, that EEG and behavioural data were not recorded simultaneously; therefore, this null finding should be treated with caution.

## Discussion

We set out to investigate the effects of expectation on stimulus decoding during statistical learning and demonstrated dynamic differences in decoding expected vs. unexpected stimulus categories, both within and across trials. EEG-based decoding of expected vs. unexpected visual scene categories showed that within trials, a relative decoding boost for expected stimuli was followed by a relative decoding boost for unexpected stimuli, in line with the recent OPT (*González-García and He, 2021*). However, across trials, the decoding boost for unexpected stimuli was observed earlier than the decoding boost for expected stimuli, possibly due to a dynamic relationship between cortical processing hierarchies (*Ahissar and Hochstein, 2004*; *Nigam and Schwiedrzik, 2024*; *Thomas et al., 2024*).

The OPT (*Press et al., 2020*) combines both sharpening and dampening models of stimulus expectation, which have previously been mapped onto a relative decoding boost for expected vs. unexpected stimuli respectively (*Han et al., 2019*). In this model, hypothesis units (units representing the brain's 'best guess' about the outside world) are initially strongly weighted towards the expected; however, when agents encounter events that elicit surprise, increased gain on surprising inputs leads to high-fidelity representations of unexpected events across hypothesis units. In simple terms, early sharpening (which favours processing of expected information) is followed by later dampening (which in turn allows novel information to be processed) within trials. This is supported by our within-trial results, whereby we found that decoding of expected images was significantly higher than that of unexpected images early during the trial, while the opposite effect was found later during the trial. Differences in early vs. late prediction error detection have previously been observed using fMRI (*Malekshahi et al., 2016*). This study used a visual detection task to demonstrate that immediate implicit (early) detection of PEs enables fast but partial comparison of bottom-up sensory input with top-down predictive information, but effortful explicit (late) processing permits more comprehensive assessment of the type of mismatch between expected and actual input. While the authors did not interpret their results in terms of dampening or sharpening models, it is possible that the differences observed are a result of changing temporal dynamics in the neural response which are difficult to demonstrate using fMRI. In this context, higher-level processes would modulate detection of prediction error through top-down processes and regulate adaptation of predictive models as a result.

Interestingly, the early (~150 ms) decoding accuracy boost found for expected trials coincided with the univariate ERP amplitude increase found for unexpected trials. Such seemingly opposing effects are commonly explained based on the notion that sharpening leads to fewer units responding to the expected stimulus, and the population response becoming sparser and more selective (*Kremláček et al., 2016*). According to Friston's original theoretical model assessment of predictive coding (*Friston, 2005*), sharpening and dampening occur in parallel but in separate prediction and error neurons which would reside in deep and superficial layers of the brain respectively. As such, studies that have argued for sharpening models (*Kok et al., 2012*; *Yon et al., 2018*) have focussed primarily on the response to expectations in the occipital cortex and V1, while studies that have favoured

dampening models (*Richter et al., 2018*; *Richter et al., 2022*; *Han et al., 2019*) have centred more on the neural response to violated expectations in the ventral visual stream. Using fMRI, *Thomas et al., 2024* found that expected and unexpected events are represented across the cortical column in V1. However, while expected events were represented similarly across layers, unexpected events were only well represented in superficial layers. One recent EEG study has attempted to directly elucidate the mechanisms of the OPT. *Xu et al., 2021* used auditory cues followed by a predictable series of flashes to investigate the mechanisms of ES. In line with OPT, they found that the N1 was augmented, while the N2 was attenuated. However, this study focused entirely on ERP analyses (rather than multivariate decoding or gain/tuning models), which limits the interpretability of the results in terms of sharpening vs. dampening. Other studies have also found evidence for dampening after learning only. For example, two studies (*Kumar et al., 2017*; *Meyer and Olson, 2011*) examining the inferior temporal (IT) cortex in monkeys using single- and multi-unit recordings have found that IT neurons represent less accurately predictable vs. unpredictable stimuli. Similarly, MEG studies have shown that representations of expected stimuli in the neural signal appear shortly before stimulus onset (*Kok et al., 2017*) and that prior expectations lead to overall dampened activity in the auditory cortex (*Han et al., 2019*). An fMRI study also showed overall dampened activity in the ventral visual stream (*Richter et al., 2018*). It is worth noting that several of the studies which showed sharpening found sharpening for predictions which were implemented as task-relevant (*Kok et al., 2012*; *Yon et al., 2018*), while some of the studies which showed dampening found effects in response to task-irrelevant predictions (*Richter et al., 2018*; *Richter et al., 2022*). However, it is unlikely that effects of task-relevance are being parsed out in the present study, where no instructions regarding relevance were given to participants, and behavioural tests were only administered after implicit learning. None of the aforementioned studies showed a reversal of sharpening and dampening effects, but it is worth noting that their experimental design did not allow for testing these effects over the course of learning as they implemented extensive training and/or exposure paradigms. Some studies did use categorised stimuli to reduce this impact, but exemplars were still repeated multiple times over the course of the experiment (*Meyer and Olson, 2011*; *Sherman et al., 2022*), or ensured that stimuli were equally familiar across expectation conditions (*Kumar et al., 2017*; *Han et al., 2019*). However, these studies given that failing to fully account for the confounding effects of RS as adaptation effects can depend on stimulus history beyond the most recently encountered stimulus and long-lag repetition effects can occur when the first and second presentation of a stimulus is separated by hundreds of intervening stimuli (*Feuerriegel et al., 2021*), which may mask the ES effects being examined. The present study controls for confounding RS effects by using stimuli that are novel and unfamiliar, foregoes a training session to examine ES effects both during learning and after learning has occurred, and utilises the high temporal resolution of EEG, providing a unique insight into the neural dynamics of these phenomena, albeit without a correlation between neural and perceptual variables.

In contrast to our within-trial results, when we examined how this process developed over multiple trials, we found that dampening occurred within the first ~15 minutes of recording and was only later followed by sharpening. These effects seem to be driven by changes in responses to unexpected trials, while expected trials did not change reliably over time. This apparent stability suggests that expectation-related representations are less sensitive to the ongoing learning-related changes that affect unexpected trials. We hypothesise that this is a result of a dynamic relationship between hierarchical levels of cortical processing. Previous work has shown that in MEG (*Cichy et al., 2014*), early-latency responses typically reflect early cortical stages, while late-latency responses typically reflect higher cortical stages. Our results are in line with this in that across trials, late-latency decoding differences arise within the initial trials, suggesting that higher-order regions quickly process statistical regularities during learning. Similarly, early-latency decoding differences occur later during the experiment, consistent with the theory that lower-order regions need more time to accumulate evidence, while the distinction between higher-order dampening and lower-order sharpening can also be observed in rapid forms of learning, as seen in rapid one-shot perceptual learning (*González-García and He, 2021*). In short, 'invalid' trials are more salient early in the experiment as priors are still being formed, which leads to predictions occurring earlier in each trial, later in the experiment. Earlier studies have pointed to a dissociation between early and late ventral stream effects, which may also fit into this framework (*Richter et al., 2018*; *Richter et al., 2022*). Other studies using statistical learning have indicated slower dynamics, in that some regions, such as the hippocampus, begin

preferentially representing PEs early during learning, and predictions are preferentially represented later during learning (*Aitken and Kok, 2022*). This is consistent with our finding that early dampening is followed by later sharpening over time, given that sharpening and dampening are defined by the relative difference in decoding accuracy between expected and unexpected trials. Another recent fMRI study found that in the presence of predictions, early stages of the processing hierarchy exhibit well-separable and high-dimensional neural geometries resembling those at the top of the hierarchy, which is accompanied by a systematic shift in tuning properties from higher to lower areas, enriching lower areas with higher-order, invariant representations in place of their feedforward tuning properties (*Riels et al., 2022*). In the context of these studies, our results could be explained by higher-order regions quickly detecting stimulus statistics and showing evidence of predictive dampening to expected inputs, and then gradually delegating those predictions to lower-order regions which start showing evidence of predictive sharpening. This suggestion is somewhat speculative, however, and is thus intended to stimulate further research. It should also be noted that we demonstrate these across-trial dynamics in the visual domain only, and other domains (such as action and audition) may elicit different results and still require additional research.

In addition to the dynamic effects of expectation on sensory decoding of the trailing image, we also found that valid expectation increases decoding accuracy of the memory of the leading image. Within trials, this effect was found at similar latencies as the positive effect of expectation on sensory decoding; however, across trials, the effect on memory decoding started earlier and persisted throughout most of the experiment. A memory trace linking the trailing and leading image is a key component of associative learning (*Libby and Buschman, 2021*). Furthermore, the finding that within a trial, sensory and memory decoding occur at similar latencies and are subject to similar effects of expectation is broadly consistent with work showing simultaneous sensory, mnemonic, and predictive representations within the same neocortical regions (*Cappotto et al., 2022*; *Barron et al., 2020*). However, the differential dynamics of these effects across trials suggest that memory and prediction encoding is also partly dissociable, not only at the level of the hypothesised circuit mechanisms, likely involving the neocortex and the hippocampus (*Jiang and Rao, 2024*), but also over time. This suggestion is consistent with a neuroimaging study showing that during associative learning, the hippocampus gradually switches from signalling PEs (i.e., mismatch between memory trace and actual stimulus) to signalling predictions (i.e., stimulus expectation independent of the actual stimulus) (*Aitken and Kok, 2022*).

In summary, our study is the first to show dynamic effects of expectation on stimulus processing during learning, both within and across trials. We have shown using EEG and decoding analyses that early dampening is followed by later sharpening over time, but within trials early sharpening is followed by later dampening. This provides direct evidence for the OPT, while also indicating that sharpening and dampening effects emerge at different learning stages.

## Methods

### Participants

Thirty-one healthy participants (25 females, 1 non-binary, 5 males) aged 18–35 (mean = 22.7) were recruited from the student population at Freie Universität Berlin. One participant was left-handed. Participants had no history of neurological illness, were not currently taking psychotropic medication and had normal or corrected-to-normal vision. The study was approved by the Ethics Committee of the Department of Education and Psychology at Freie Universität Berlin (038.2023). Participants were compensated with either €30 or research participation credits. Data from one participant was excluded due to a technical issue which resulted in an absence of triggers in the EEG data. The final sample size (N=30) corresponded to previous research (*Huang et al., 2024*).

### Experimental design and statistical analysis

Stimuli and experimental paradigm

Main task

Approximately 3500 colour images across nine categories were generated using artificial intelligence software Craiyon (Craiyon LLC, 2022). Images were visually inspected to ensure they appeared realistic. Some physically implausible examples were discarded. Stimuli were presented on a 38 cm LCD monitor with 1280 × 1024 resolution and 60 Hz refresh rate, the viewing distance was 62 cm,

and stimuli were presented at 2.55°. The experimental paradigm was adapted from *Richter et al., 2018* with several modifications, as described below. Participants were exposed to pairs of visual stimuli, comprising artificially generated natural scenes drawn from different scene categories. In each trial, participants were presented with two images from different categories in quick succession, each presented for 100 ms with 800 ms interstimulus interval and 1300–2200 ms intertrial interval (*Figure 1A*). A fixation cross was presented throughout the experiment. Each image belonged to one of nine scene categories, with five 'Leading' categories ('barn', 'beach', 'library', 'restaurant', 'cave') and four 'Trailing' categories ('church', 'conference room', 'castle', 'forest'). Category pairs and the transitional probabilities between them were determined by the transitional probability matrix depicted in *Figure 1B*. As previously established, many prior studies have overlooked the confounding effects of RS in their experimental paradigms. To control for RS, categorised images were used to avoid the repetition of stimuli. As such, each individual image was only presented once. In the experimental condition, categories were paired in a 2:1 manner, where two different 'Leading' categories could result in one 'Trailing' category with 75% validity. For example, both 'beach' and 'barn' as 'Leading' categories would result in 'church' as a 'Trailing' category with 75% validity. In the control condition, one 'Leading' category led to two 'Trailing' categories with 50%/50% accuracy. Participants were not informed of the associations between categories and were instead instructed to respond by button press when images appeared upside down, which occurred in ~5% of trials. This acted as catch trials to ensure participants maintained attention on the images. The occurrence of these catch trials was randomised, as was trial order. Participants did not undergo a training session, such that the neural signatures of early learning processes may be investigated using EEG. Each participant completed eight blocks consisting of 216 trials per block, totalling ~90 minutes of testing.

### Categorisation task

EEG data was collected from the first 10 participants as a pilot version of the experiment, after which the categorisation task was added to the experiment, resulting in a lower number of participants with behavioural results. After completion of the main task, the majority of participants (n=21) performed a categorisation task to ensure learning had taken place. In this task, participants were presented with the same images as before and instructed to indicate via button press whether the 'Trailing' image takes place indoors or outdoors. This task aimed to assess implicit RT, as the statistical regularities learned in the main task could be used to predict what category the 'Trailing' image would belong to. Participants were not informed of the intent behind this task or instructed to make use of what they had learned in the main task.

## Data analysis

### Categorisation task analysis

Behavioural data from the categorisation task was analysed in terms of RTs. All RTs exceeding 2 SD above or below the mean across subjects were excluded as outliers. RTs for valid, invalid and neutral trials underwent a log transformation before being averaged separately per participant. A repeated measures ANOVA was performed to compare RTs in each condition, then three separate paired t-tests were conducted comparing valid vs. invalid, valid vs. neutral and invalid vs. neutral. Removal of RTs exceeding 2 SDs above or below the mean resulted in two participants having no neutral trials, thus these empty values were replaced with the mean of the remaining subjects. In order to examine the extent to which participants' neural responses were linked to their behavioural responses, we calculated Pearson correlation coefficients for behavioural responses, neural responses during the sharpening effect (123–180 ms), and neural responses during the dampening effect (280–296 ms) based on the difference in mean decoding across time within both time windows for each participant.

### EEG data preprocessing

EEG data were acquired at 2048 Hz from 64 electrodes using a BioSemi ActiveTwo system. Electrodes were arranged according to the international 10/20 system. Preprocessing was conducted using custom SPM12 (Wellcome Trust Centre for Neuroimaging, University College London; RRID:SCR_007037) for Matlab (The MathWorks; RRID:SCR_001622) as well as custom Matlab scripts. Continuous EEG data were high-pass filtered above 0.1 Hz and notch-filtered between 48–52 Hz using fifth-order zero-phase Butterworth filters. Data were then downsampled to 300 Hz. Blinks were

detected using two horizontal and two vertical electro-oculogram (EOG) electrodes placed above and below the left eye, and to the left and right of the eyes. An epoch of –200 ms to 400 ms relative to artefact onset was applied to detected eye blinks and spatiotemporal confounds were calculated using singular value decomposition (SVD). Top two principal components were removed from the segments of data associated with each eye blink (*Ille et al., 2002*). Data were then further denoised using the Dynamic Separation of Sources (*de Cheveigné and Simon, 2008*) which maximises the reproducibility of stimulus-evoked responses across trials in order to increase the signal-to-noise ratio of ERPs. During this process, data were re-referenced offline to the average of all channels. Continuous data were epoched between 200 ms before leading stimulus onset to 2600 ms after stimulus onset. Each epoch was baseline-corrected to the mean of the pre-stimulus onset (from –200 ms to 0 ms relative to stimulus onset). Epochs with an average root mean square (RMS) amplitude exceeding the median by 2 SDs (SDs) were excluded from analysis in order to remove contamination by transient artefacts. This resulted in rejecting 11.2% (±1.03% the standard error of the mean (SEM) across participants) trials on average. Data were then averaged across trials and a low-pass filter of 48 Hz was applied. Catch trials were removed prior to analysis.

## Decoding analyses

Single-trial data underwent decoding based on linear support vector machines (SVM) implemented using custom MATLAB code. As part of feature selection, principal component analysis (PCA) was conducted to reduce dimensionality of sensor-level EEG data, and subsequent components were selected based on signal-to-noise (SNR) with a cutoff threshold of 8 dB. As a result, an average of 7.6 components, SD = 0.498, was used across participants. Given the poor spatial resolution of EEG, considering all channels for feature extraction increases the computational cost, risk of overfitting and required number of trials. As such, PCA can be useful for dimensionality reduction prior to SVM-based decoding (*Zhang et al., 2019*; *Asadur Rahman et al., 2020*). Three separate decoding analyses were conducted, decoding the visual category of both leading and trailing images, decoding the predictable trailing images based on the leading images, and memory decoding for trailing images. Valid and invalid were analysed separately at each step. In both analyses (of valid vs. invalid trials), the decoder was trained and tested using leave-8-out cross-validation, such that in each test set, single trials corresponding to all 8 unique combinations of 4 leading and 2 trailing images were included (see *Figure 1B*). Visual category decoding was used to differentiate neural responses to leading vs. trailing categories, while memory decoding aimed to assess memory of the leading category based on the trailing category.

The initial analysis aimed at quantifying decoding as a function of peristimulus time. To this end, we calculated the decoding accuracy per time point, using all trials. The resulting decoding time series were converted to one-dimensional images, entered into a GLM, outliers larger than 3 SDs based on Cook's distance were removed, and subjected to *Friston, 2005* a one-sample t-test against chance-level, equivalent to a fixed-effects analysis (*Allefeld et al., 2016*), and (*Jiang and Rao, 2024*) a paired t-test between valid and invalid categories. While the fixed effects approach (one-sample t-test) enables us to establish whether some consistent informative patterns are detectable in these particular subjects, the results from our paired t-tests support inference to the wider population. Given that EEG time-series are autocorrelated over time, statistical tests were corrected for multiple comparisons over time using cluster-based FWE correction. A follow-up analysis aimed at quantifying changes in decoding over learning. To this end, based on the time windows in which we found significant decoding differences between valid and invalid trials (see below), we averaged the decoding accuracy estimates within each time window, separately for four bins of trials. In order to assess the utility of examining each bin in more detail, we calculated the trial-by-trial time-series of the decoding accuracy benefit for valid vs. invalid for each participant and averaged this benefit across time points for each of the two significant time windows, resulting in 200 trials per participant. Based on this, we fitted a logarithmic model to quantify the change of this benefit over trials, then found the trial index for which the change of the logarithmic fit was <0.1%. Logarithmic fits have been shown to accurately describe learning curves in statistical learning (*Kang et al., 2023*; *Siegelman et al., 2018*; *Choi et al., 2020*), informing our choice of model. In the early time window (123–180 ms), the median proportion of trials after which the decoding benefit stabilises is 11%, with the 99th percentile being reached after 38% of trials. In the later time window (280–196 ms), the median proportion of trials after which

the decoding benefit stabilises is 18%, with the 99th percentile being reached after 47% of trials. Given that decoding stabilises after approximately 50% of trials (indicating that virtually zero learning took place after this point), and in order to balance this with having a sufficient number of trials for analysis, we focussed our further analyses on bins 1–2 to directly assess the effects of learning. Each bin encompassed approximately 15 minutes of testing or 432 trials on average.

# Additional information

## Funding

| Funder | Grant reference number | Author |
|---|---|---|
| Deutsche Forschungsgemeinschaft | AU423/2-1 | Ryszard Auksztulewicz |
| European Research Council | 101001270 | Federico de Martino |
| Deutsche Forschungsgemeinschaft | SCHW1683/2-1 | Caspar M Schwiedrzik |

The funders had no role in study design, data collection and interpretation, or the decision to submit the work for publication.

## Author contributions

Hannah H McDermott, Conceptualization, Data curation, Formal analysis, Investigation, Visualization, Methodology, Writing – original draft, Writing – review and editing; Federico de Martino, Conceptualization, Formal analysis, Supervision, Writing – original draft, Writing – review and editing; Caspar M Schwiedrzik, Conceptualization, Formal analysis, Funding acquisition, Writing – original draft, Writing – review and editing; Ryszard Auksztulewicz, Conceptualization, Resources, Software, Formal analysis, Supervision, Funding acquisition, Validation, Investigation, Visualization, Methodology, Writing – original draft, Project administration, Writing – review and editing

## Author ORCIDs

Hannah H McDermott ⓘD https://orcid.org/0009-0005-5399-2917
Federico de Martino ⓘD https://orcid.org/0000-0002-0352-0648
Caspar M Schwiedrzik ⓘD https://orcid.org/0000-0003-0661-8859
Ryszard Auksztulewicz ⓘD https://orcid.org/0000-0001-9078-3667

## Ethics

Ethical approval was provided by the Ethics Commission of the Department of Educational Sciences and Psychology at Freie Universität Berlin (038.2025). Informed consent was obtained from subjects prior to data collection.

Reviewer #3 (Public review): https://doi.org/10.7554/eLife.103689.4.sa1
Author response https://doi.org/10.7554/eLife.103689.4.sa2

# Additional files

## Supplementary files

MDAR checklist

## Data availability

Scripts and stimuli are available at https://github.com/hannahmcderm/Expectation_EEG_eLife (copy archived at *McDermott, 2025*) data is available at https://osf.io/x7ydf.

The following dataset was generated:

| Author(s) | Year | Dataset title | Dataset URL | Database and Identifier |
|---|---|---|---|---|
| McDermott H | 2025 | ExpectationSuppression_EEG_eLife | https://osf.io/x7ydf | Open Science Framework, x7ydf |

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
