## [Editor Report · eLife Assessment]

This **important** study is of relevance to the fields of predictive processing, perception, and learning, with a well-designed paradigm allowing the authors to avoid several common confounds in investigating predictions, such as adaptation. Using a state-of-the-art multivariate EEG approach, the authors test the opposing process theory and find evidence in support of it. Overall, the empirical evidence is **solid**; however, some conclusions rest on limited evidence and need further work to reconcile the present results with previous studies.

---

## [Referee Report · Reviewer #3 (Public review)]

Summary:

In their study McDermott et al. investigate the neurocomputational mechanism underlying sensory prediction errors. They contrast two accounts: representational sharpening and dampening. Representational sharpening suggests that predictions increase the fidelity of the neural representations of expected inputs, while representational dampening suggests the opposite (decreased fidelity for expected stimuli). The authors performed decoding analyses on EEG data, showing that first expected stimuli could be better decoded (sharpening), followed by a reversal during later response windows where unexpected inputs could be better decoded (dampening). These results are interpreted in the context of opposing process theory (OPT), which suggests that such a reversal would support perception to be both veridical (i.e., initial sharpening to increase the accuracy of perception) and informative (i.e., later dampening to highlight surprising, but informative inputs).

Strengths:

The topic of the present study is of significant relevance for the field of predictive processing. The experimental paradigm used by McDermott et al. is well designed, allowing the authors to avoid common confounds in investigating predictions, such as stimulus familiarity and adaptation. The introduction provides a well written summary of the main arguments for the two accounts of interest (sharpening and dampening), as well as OPT. Overall, the manuscript serves as a good overview of the current state of the field.

Weaknesses:

In my opinion the study has a few weaknesses. Some method choices appear arbitrary (e.g., binning). Additionally, not all results are necessarily predicted by OPT. Finally, results are challenging to reconcile with previous studies. For example, while I agree with the authors that stimulus familiarity is a clear difference compared to previous designs, without a convincing explanation why this would produce the observed pattern of results, I find the account somewhat unsatisfying.

---

## [Author Response]

The following is the authors’ response to the previous reviews

**Reviewer 1**
MinorThe main substance of my previous comment I suppose targeted a deeper issue - namely whether such a result is reflecting a resolution to a 'neural prediction' puzzle or a 'perceptual prediction' puzzle. Of course, these results tell us a great deal about a potential resolution for how dampening and sharpening might co-exist in the brain - but in the absence of corresponding perceptual effects (or a lack of correlation between neural and perceptual variables - as outlined in this revision) I do wonder if any claims about implications for perception might need moderation or caveating. To be honest, I don't think the authors *need* to make any more changes along these lines for this paper to be acceptable - it is more an issue they might wish to consider themselves when contextualizing their findings.

Thank you for the thoughtful comment. We have now added a caveat to the relevant section of the discussion to make it clearer that we are discussing neural results, not perceptual results (p.20, lines 378-379).

I am also happy with the changes that the authors have made justifying which claims can and cannot made based on a statistical decoding test against 'chance' in a single condition using t-tests. I was perhaps a little unclear when I spoke about 'comparisons against 0' in my original review, when the key issue (as the authors have intuited!) is about comparisons against 'chance' (where e.g., 0% decoding above chance is the same thing as 'chance'!). The authors are of course correct in the amendment they have made on p.29 to make clear this is a 'fixed effects analysis' - though I still worry this could be a little cryptic for the average reader. I am not suggesting that the authors run more analyses, or revise any conclusions, but I think it would be more transparent if a note was added along the lines of "while the fixed effects approach (one-sample t-test) enables us to establish whether some consistent informative patterns are detectable in these particular subjects, the results from our paired t-tests support inference to the wider population".

This sentence has been added for increased transparency (p. 27, lines 544-547).

**Reviewer 3**
Major(1) In the previous round of comments, I noted that: "I am not fully convinced that Figures 3A/B and the associated results support the idea that early learning stages result in dampening and later stages in sharpening. The inference made requires, in my opinion, not only a significant effect in one-time bin and the absence of an effect in other bins. Instead to reliably make this inference one would need a contrast showing a difference in decoding accuracy between bins, or ideally an analysis not contingent on seemingly arbitrary binning of data, but a decrease (or increase) in the slope of the decoding accuracy across trials. Moreover, the decoding analyses seem to be at the edge of SNR, hence making any interpretation that depends on the absence of an effect in some bins yet more problematic and implausible". The authors responded: "we fitted a logarithmic model to quantify the change of the decoding benefit over trials, then found the trial index for which the change of the logarithmic fit was < 0.1%. Given the results of this analysis and to ensure a sufficient number of trials, we focused our further analyses on bins 1-2". However, I do not see how this new analysis addresses the concern that the conclusion highlights differences in decoding performance between bins 1 and 2, yet no contrast between these bins are performed. While I appreciate the addition of the new model, in my current understanding it does not solve the problem I raised. I still believe that if the authors wish to conclude that an effect differs between two bins they must contrast these directly and/or use a different appropriate analysis approach.Relatedly, the logarithmic model fitting and how it justifies the focus on analysis bin 1-2 needs to be explained better, especially the rationale of the analysis, the choice of parameters (e.g., why logarithmic, why change of logarithmic fit < 0.1% as criterion, etc), and why certain inferences follow from this analysis. Also, the reporting of the associated results seems rather sparse in the current iteration of the manuscript.

We thank the reviewer for this important point. Following your suggestion, we conducted additional post-hoc tests directly comparing the first and second bins. We found significant differences between bins in the invalid trials, but not the valid trials, suggesting that sharpening/dampening effects are condition specific. This is discussed in the manuscript on p.14, lines 268-271; p.15, 280-284; p.20, lines 382-386.

A logarithmic analysis was chosen as learning is usually found to be a nonlinear process; learning effects occur rapidly before stabilising relatively early, as seen in Fig. 2D. This is consistent with other research which found that logarithmic fits efficiently describe learning curves in statistical learning (Kang et al., 2023; Siegelman et al., 2018; Choi et al., 2020). By utilising a change of logarithmic fit at <0.1% as a criterion, it is ensured that virtually zero learning took place after that point, allowing us to focus our analysis on learning effects as they developed and providing a more accurate model of representational change. This is explained in the manuscript on p.13, lines 250-251; p.27-28, lines 557-563.

(2) A critical point the authors raise is that they investigate the buildup of expectations during training. They go on to show that the dampening effect disappears quickly, concluding: "the decoding benefit of invalid predictions [...] disappeared after approximately 15 minutes (or 50 trials per condition)". Maybe the authors can correct me, but my best understanding is as follows: Each bin has 50 trials per condition. The 2:1 condition has 4 leading images, this would mean ~12 trials per leading stimulus, 25% of which are unexpected, so ~9 expected trials per pair. Bin 1 represents the first time the participants see the associations. Therefore, the conclusion is that participants learn the associations so rapidly that ~9 expected trials per pair suffice to not only learn the expectations (in a probabilistic context) but learn them sufficiently well such that they result in a significant decoding difference in that same bin. If so, this would seem surprisingly fast, given that participants learn by means of incidental statistical learning (i.e. they were not informed about the statistical regularities). I acknowledge that we do not know how quickly the dampening/sharpening effects develop, however surprising results should be accompanied with a critical evaluation and exceptionally strong evidence (see point 1). Consider for example the following alternative account to explain these results. Category pairs were fixed across and within participants,i.e. the same leading image categories always predicted the same trailing image categories for all participants. Some category pairings will necessarily result in a larger representational overlap (i.e., visual similarity, etc.) and hence differences in decoding accuracy due to adaptation and related effects. For example, house barn will result in a different decoding performance compared to coffee cup barn, simply due to the larger visual and semantic similarity between house and barn compared to coffee cup and barn. These effects should occur upon first stimulus presentation, independent of statistical learning, and may attenuate over time e.g., due to increasing familiarity with the categories (i.e., an overall attenuation leading to smaller between condition differences) or pairs.

We apologise for the confusion, there are 50 expected trials per bin per condition. The trial breakdown is as follows. Each participant completed 1728 trials, split equally across 3 mappings (two 2:1 maps and one 1:2 map), giving 1152 trials in the 2:1 mapping. Stimuli were expected in 75% of trials (864), leaving 216 per bin, and 54 per leading image in each bin. We have clarified this in the script (p.14, line 267; p.15, line 280). This is in line with similar studies in the field (e.g. Han et al., 2019).

(3) In response to my previous comment, why the authors think their study may have found different results compared to multiple previous studies (e.g. Han et al., 2019; Kumar et al., 2017; Meyer and Olson, 2011), particularly the sharpening to dampening switch, the authors emphasize the use of non-repeated stimuli (no repetition suppression and no familiarity confound) in their design. However, I fail to see how familiarity or RS could account for the absence ofsharpening/dampening inversion in previous studies.First, if the authors argument is about stimulus novelty and familiarity as described by Feuerriegel et al., 2021, I believe this point does not apply to the cited studies. Feuerriegel et al., 2021 note: "Relative stimulus novelty can be an important confound in situations where expected stimulus identities are presented often within an experiment, but neutral or surprising stimuli are presented only rarely", which indeed is a critical confound. However, none of the studies (Han et al., 2019; Richter et al., 2018; Kumar et al., 2017; Meyer and Olson, 2011) contained this confound, because all stimuli served as expected and unexpected stimuli, with the expectation status solely determined by the preceding cue. Thus, participants were equally familiar with the images across expectation conditions.Second, for a similar reason the authors argument for RS accounting for the different results does not hold either in my opinion. Again, as Feuerriegel et al. 2021 correctly point out: "Adaptation-related effects can mimic ES when the expected stimuli are a repetition of the last-seen stimulus or have been encountered more recently than stimuli in neutral expectation conditions." However, it is critical to consider the precise design of previous studies. Taking again the example of Han et al., 2019; Kumar et al., 2017; Meyer and Olson, 2011. To my knowledge none of these studies contained manipulations that would result in a more frequent or recent repetition of any specific stimulus in the expected compared to unexpected condition. The crucial manipulation in all these previous studies is not that a single stimulus or stimulus feature (which could be subject to familiarity or RS) determines the expectation status, but rather the transitional probability (i.e. cue-stimulus pairing) of a particular stimulus given the cue. Therefore, unless I am missing something critical, simple RS seems unlikely to differ between expectation condition in the previous studies and hence seems implausible to account for differences in results compared to the current study.Moreover, studies cited by the authors (e.g. Todorovic & de Lange, 2012) showed that RS and ES are separable in time, again making me wonder how avoiding stimulus repetition should account for the difference in the present study compared to previous ones. I am happy to be corrected in my understanding, but with the currently provided arguments by the authors I do not see how RS and familiarity can account for the discrepancy in results.

The reviewer is correct in that the studies cited (Han et al., 2019; Kumar et al., 2017; Meyer and Olson, 2011) ensure that participants are equally familiar with the images across expectation conditions. Where the present study differs is that participants are not familiar with individual exemplars at all. Han et al., 2019 used a pool of 30 individual images, and subjects underwent exposure sessions lasting two hours each daily for 34 days prior to testing. Kumar et al., 2017 used a pool of 12 images with subjects being exposed to each sequential pair 816 times over the course of the training period. Meyer & Olsen, 2011 used pure tones at five different pitch levels. While familiarity of stimuli across conditions was controlled for in these studies in the sense that familiarity was constant across conditions, novelty was not controlled for. The present study uses a pool of ~3500 images, which are unrepeated across trials.

Feuerriegel et al., 2021 also points out: “There are also effects of adaptation that are dependent on the recent stimulation history extending beyond the last encountered stimulus and long-lag repetition effects that occur when the first and second presentation of a stimulus is separated by tens or even hundreds of intervening images”. Bearing this in mind, and given the very small pool of stimuli being used by Han et al., 2019; Kumar et al., 2017; Meyer and Olson, 2011, it stands to reason that these studies may still have built-in but unaccounted for effects relating to the repetition of exemplars. Thus, our avoidance of those possible confounds, in addition to foregoing any prior training, may elicit differing results. Furthermore, as pointed out by Walsh et al. 2020, methodological heterogeneity (such as subject training) can produce contrasting results as PP makes divergent predictions regarding the properties of prediction error given different permutations of variables such as training, transitional probabilities, and conditional probabilities. In our case, the use of differing methodology was intentional. These issues have been discussed in more detail on p.5, lines 112-115; p.19, lines 368-377; p.20, lines 378-379.

Minor(1) The authors note in their reply to my previous questions that: "As mentioned above, we opted to target our ERP analyses on Oz due to controversies in the literature regarding univariate effects of ES (Feuerriegel et al., 2021)". This might be a lack of understanding on my side, but how are concerns about the reliability of ES, as outlined by Feuerriegel et al. (2021), an argument for restricting analyses to 1 EEG channel (Oz)? Could one not argue equally well that precisely because of these concerns we should be less selective and instead average across multiple (occipital) channels to improve the reliability of results?

The reviewer is correct in suggesting that a cluster of occipital electrodes may be more reliable than reporting one single electrode. We have amended the analysis to examine electrodes Oz, O1, and O2 (p.9, lines 187-188; p.11, lines 197-201).

(2) The authors provide a github link for the dataset and code. However, I doubt that github is a suitable location to share EEG data (which at present I also cannot find linked in the github repo). Do the authors plan to share the EEG data and if so where?

Thank you for bringing this to my attention. EEG data has now been uploaded at osf.io/x7ydf and linked to the github repository (p.28, lines 569-570).

(3) The figure text could benefit from additional information; e.g. Fig.1C and Fig.3 do not clarify what the asterisk indicates; p < ? with or without multiple comparison correction?

Thank you for pointing out this oversight, the figure texts have been amended (p. 9, line 168; p.16, line 289).